# Developmental Toxicities in Zebrafish Embryos Exposed to Tri-o-cresyl Phosphate

## Meng Li, Congcong Wang, Wanying Gui, Peng Wang, Jierong Chen, Shaoqi Zuo, Yanbin Zhao, Jiayin Dai and Kun Zhang *

State Environmental Protection Key Laboratory of Environmental Health Impact Assessment of Emerging Contaminants, School of Environmental Science and Engineering, Shanghai Jiao Tong University, 800 Dongchuan Road, Shanghai 200240, China; limeng0038@sjtu.edu.cn (M.L.); wcongcong321@126.com (C.W.); guiwanying@ioz.ac.cn (W.G.); 11849364@mail.sustech.edu.cn (P.W.); ali-chenjierong@sjtu.edu.cn (J.C.); zuoshaoqi@sjtu.edu.cn (S.Z.); zhaoyanbin@sjtu.edu.cn (Y.Z.); daijy65@sjtu.edu.cn (J.D.)
* Correspondence: kunzhang@sjtu.edu.cn; Tel.: +86-1881-8207-712

**Abstract:** As a widely used plasticizer and fire retardant, tri-o-cresyl phosphate has been commonly found in global water sources, sediments and biota. However, its potential toxicity to aquatic organisms is not fully understood. Here, we assessed its developmental effects by use of a zebrafish (*Danio rerio*) model at tri-o-cresyl phosphate concentrations between 0.15 and 88.5 μg/L. Diverse impairments of zebrafish embryos, such as altered morphological and physical characteristics and locomotor behaviors, were observed at different tri-o-cresyl phosphate concentrations. Furthermore, swimming behaviors were significantly inhibited at tri-o-cresyl phosphate concentrations ranging from 3.0 μg/L to 88.5 μg/L. The swimming activity during light-to-dark transition significantly increased at tri-o-cresyl phosphate concentrations of 14.5 μg/L to 88.5 μg/L. Taken together, our present data help to clarify the potential developmental toxicity of tri-o-cresyl phosphate that was not yet fully recognized, and thus contribute to its environmental risk assessment.

**Keywords:** organophosphorus flame retardants; aquatic organism; toxicity; morphological effect; swimming behavior





## 1. Introduction

With the regulations from European REACH (Registration, Evaluation and Authorization of Chemicals) and the Stockholm Convention, polybrominated diphenyl ethers (PBDEs), a kind of widely employed flame retardant, have been prohibited due to their harmful characteristics of persistence, long-range transportability, bioaccumulation and biological toxicity [1,2]. Meanwhile, as a novel type of substitute, organophosphorus flame retardants (OPFRs) are widely applied in various consumer and industrial material products, such as electronic equipment, plastics, textiles, building materials and antifoaming agents [3]. In the year of 1992, global consumption of OPFRs was only 100,000 tons [4]. However, the yield of OPFRs sharply increased to 680,000 tons in 2015 [5,6]. Until now, there have been at least 360 factories producing OPFR monomers and OPFR mixtures globally. Of them, it is worth mentioning that there were more than 200 factories located in mainland China [6].

Among OPFRs, tricresyl phosphate (TCP) emerged as a representative component. It is composed of three major isomers, including tri-o-cresyl phosphate (ToCP), tri-m-cresyl phosphate (TmCP) and tri-p-cresyl phosphate (TpCP), with a yield of at least 11,000 tons per year [7,8]. As a consequence, residues of TCP were widely detected in surface water, sediment, indoor dust and biota. In Japan, TCP concentrations were found in domestic wastewater at levels of up to 0.56 μg/L, of which ToCP was the predominant proportion (contributed nearly 52%) [9]. In surface water, TCP was detected at up to 0.05 μg/L in SongHua River, China [10], and in drinking water up to 0.11 μg/L in Republic of Korea [11].

In addition, TCP can be enriched in the sedimentary phase. Residues of TCP in sediments of the Lian River, China have been found at 17.0 µg/g dw [12]. TCP concentrations in the indoor dust of urban homes of Guangzhou, China were up to 7.74 µg/g, and in the indoor dust of rural e-waste workshops of Qingyuan, China could be up to 46.6 µg/g [13]. In a special case, TCP was detected as high as 130 µg/g in the soil samples collected at US air force bases with hydraulic fluids pollution [14]. TCP was even found in human breast milk in Japan, the Philippines and Vietnam, with concentrations of up to 85 ng/g [15].

The widespread occurrences of TCP in aquatic systems raise concerns about its potential risks to aquatic organisms, especially to fish. In vivo and in vitro studies have demonstrated that TCP exerts neurotoxic effects and liver damage and impacts on the endocrine and reproductive systems of fish. For instance, TCP exposure caused significant dilation of the efferent duct in the testes and fertilization inhibition of Japanese medaka at concentrations of 0.66 µg/L and above [16]. After subjecting adult zebrafish to a 14-day exposure to TCP, significant alterations in plasma hormone concentrations were observed. Among female fish, there were remarkable increases in plasma testosterone (T) and estradiol ($E_2$) levels, while 11-ketotestosterone (11-KT) levels remained unchanged. Conversely, male zebrafish showed a decrease in both T and 11-KT levels but exhibited an increase in $E_2$ concentration. Moreover, significant changes in the transcriptions of key genes involved in hormone biosynthesis, such as CYP17 and CYP19a, were further observed at TCP concentrations of 0.01 µg/L and above [17].

Among the isomers of TCP, tri-o-cresyl phosphate (ToCP) was reported as by far the most toxic one in acute and short-term exposures. For instance, it is the only isomer known to induce delayed neurotoxicity. Studies have shown that a single oral dose of 50–500 mg/kg of ToCP can lead to delayed neuropathy in chickens, whereas doses of 840 mg/kg or more were necessary to produce spinal cord degeneration in Long–Evans rats [18]. Furthermore, ToCP has been found to have significant reproductive toxicity, particularly affecting the male reproductive system. It can cause histopathological damage in the testes, leading to reduced sperm motility and sperm number. ToCP has also been associated with autophagy of the spermatogonial stem cells, further highlighting its adverse effects on male fertility [19,20]. Besides, ToCP has been recognized as a potent ligand of estrogen receptor alpha (ERα), which can lead to the activation of genes associated with tumor growth, invasion and metastasis [21]. In teleost, recent studies have demonstrated that ToCP can alter the locomotor activity of zebrafish at concentrations as low as 2541 µg/L and induce neurobehavioral alterations at concentrations of 442 µg/L [22]. ToCP has also been shown to inhibit neuroprogenitor cell proliferation and increase the incidence of abnormal zebrafish embryos at concentrations of several µM and above [23]. Despite the increasing knowledge, a thorough understanding of the potential environmental risks of ToCP on aquatic organisms is still lacking.

Thus, in the present study, we aimed to evaluate the developmental effects of ToCP, one of the major isomers of TCP, by use of zebrafish model. Multiple developmental parameters of zebrafish embryos, including yolk extension formation and spontaneous contraction, as well as their growth, cardiac functions and swimming behaviors, were assessed. The objective of this study was to advance our understanding of the impacts of OPFRs on the embryonic development of teleost and to offer new insights into their potential environmental risks.

## 2. Materials and Methods

**Chemicals.** ToCP (CAS: 78-30-8, purity ≥ 97%) was purchased from ANPEL Laboratory Technologies (Shanghai, China). Dimethylsulfoxide (DMSO) (purity ≥ 99.9%) was obtained from Macklin (Shanghai, China). HPLC/UHPLC-UV grade acetonitrile (ACN) and water were purchased from Fisher Scientific (Fair Lawn, NJ, USA). LCMS/HPLC grade formic acid was purchased from Anaqua Chemicals Supply (Wilmington, DE, USA).

**Maintenance of Zebrafish.** Zebrafish care and maintenance was performed as described previously [24,25]. In brief, the adult zebrafish (AB strain) used in the experiment

were obtained from the China Zebrafish Resource Center (Wuhan, China). Fish were maintained in a recirculating aquaculture system (ESEN, Beijing, China) at the temperature of $28 \pm 1$ °C, with a conductivity of 500–550 μS/cm and pH value of 6.8–7.5. The photoperiod was 14:10 h light/dark. They were fed twice a day with newly hatched brine shrimp. Eggs were obtained by pairwise mating of zebrafish in spawning boxes in the early morning just after fertilization. Then, they were washed twice and incubated at 28 °C in standard $E_3$ medium, which contained 5 mM NaCl, 0.17 mM KCl, 0.33 mM $MgSO_4$ and 0.33 mM $CaCl_2$.

**Experimental design.** Thirty zebrafish embryos at approximately 2 h post fertilization (hpf) were placed into each well of 6-well plates containing 5 mL reconstituted $E_3$ medium (5 mM NaCl, 0.33 mM $CaCl_2$, 0.33 mM $MgSO_4$ and 0.17 mM KCl) at a temperature of $28 \pm 1$ °C. The experimental setup consisted of 0.32, 1.6, 8.0, 40 and 200 μg/L ToCP exposure treatments, following by the solvent control group (0.1% DMSO). Four replicates were employed for each treatment. Every 24 h, lethal effects were evaluated, and dead embryos were removed. Water was completely changed every 24 h, with the new reconstituted $E_3$ fish water containing appropriate ToCP concentrations. Multiple developmental parameters, including yolk extension formation of the embryo at 16 hpf, spontaneous contraction at 24 hpf, heart functions at 56 hpf and deformations, were recorded [26]. At 120 hpf, zebrafish larvae were transferred into 96-well plates for swimming behavior analysis.

**Early development.** Images of yolk extension were captured via a stereomicroscope (Nikon, SMZ745T, Japan, Tokyo) for 16 hpf embryos. The videos capturing spontaneous contractions of embryos were obtained using a stereomicroscope (Nikon, SMZ745T, Japan, Tokyo). The recordings were made for a duration of 30 s, specifically focusing on 24 hpf embryos. Spontaneous contractions were evaluated by use of ImageJ software (NIH, Bethesda, MD, USA) with custom scripts [25]. In brief, zebrafish embryos were tracked according to changes in pixel intensity using the custom scripts designed. A series of .csv files were then exported automatically to capture the data. The frequencies of the spontaneous contractions of embryos were extracted from the change in pixel intensity of the embryos over time with a custom R script. The distance from the center point of the eye to the ear was detected for 36 hpf embryos, and defined as the distance from eye to ear (DEE) [27].

**Cardiac physiology.** For cardiac function analysis, individual embryo was positioned on a glass depression slide in a lateral position and acclimated to the microscope illumination for 10 s. Afterwards, a high-speed digital camera (Zyla 5.5 sCMOS, Oxford Instruments, Abingdon, UK) mounted on an inverted microscope (Nikon, Eclipse Ti2-E, Japan, Tokyo) was used for recording. Videos were acquired for 6 s at a rate of 60 frames per second. Heart rate, stroke volume, cardiac output and ventricular fractional area change (FAC) in embryos were measured with ImageJ based on the method we developed previously [25]. In brief, we utilized a simplified ellipsoid model to estimate the stroke volumes (SV) of the zebrafish ventricle. The length of the long axis was denoted as L, and the length of the short axis was represented as S. The SV was calculated using the formula: $SV = 4/3\pi \times L \times S^2$. Cardiac output (CO) was estimated by multiplying ventricular stroke volume by the heart rate ($CO = SV \times HR$). The FAC was evaluated as the difference between the end-diastolic diameter (Ld) and the end-systolic diameter ($L_s$), divided by the end-diastolic diameter ($L_d$) ($FAC = (L_d - L_s)/L_d$).

**Body length.** The data for body length of 60 hpf embryos were obtained using an inverted microscope (Nikon, Eclipse Ti2-E, Japan, Tokyo) coupled with a high-speed digital camera, Zyla 5.5 (Andor, Belfast, UK). At this stage of development, the swim bladders of the embryos were not fully developed, which caused them to remain almost stationary in the medium. This allowed for easier profiling and capturing of images. Using ImageJ software (NIH, Bethesda, MD, USA), the body length measurements were performed on the profile pictures of each group of larvae. Specifically, the average length from the head to tail was calculated.

**Swimming behavior.** Multiple swimming behavior parameters were quantified by use of EthoVision XT V17 (Noldus, Wageningen, The Netherlands), a well-established

zebrafish video tracking and analysis system [28]. In brief, twenty-four larvae at 120 hpf for each treatment (six for each replicate) were transferred to a 96-well plate and incubated at $28 \pm 1$ °C. After an initial acclimation of about one hour, the digital videos were recorded for 5 min in the phases of light and dark, respectively. Notably, swimming behaviors in the dark environment were captured using an infrared camera, ensuring minimal disturbance to the larvae during the recordings. Afterwards, the parameters, including swimming distance, swimming velocity and swimming time, as well as the swimming distance during light-dark transition, were measured according to the instructions.

**Chemical analysis.** For chemical analysis, two groups of water samples were collected at day 1 and day 3, respectively, for the control and each ToCP treatment. For each group of water samples, equal amounts of water from each replicate were pooled (16 mL in total for the control and each ToCP treatment). Additionally, water samples collected at the beginning of the daily experiment (T0) and 24 h later (T24) just before water renewal were mixed to create a composite sample that represented the mean exposure concentrations during the entire exposure period. Afterward, the water samples were centrifuged at 12,000 rpm for 15 min and the supernatants were collected for UPLC-MS/MS analysis.

The concentrations of ToCP were measured by UPLC-MS/MS with ESI operated in positive mode (ACQUITY UPLC, Xevo TQ-XS Triple Quadrupole Mass Spectrometry, Waters, Milford, MA, USA). LC mobile phase A was 90% water and 10% acetonitrile mixed with 0.1% formic acid (*v/v*). Mobile phase B was acetonitrile. The LC analytical column was ACQUITY UPLC CSH C18 column (100 mm × 2.1 mm × 1.7 μm; Waters, Milford, MA, USA). The column temperature was set at 40 °C. A gradient elution was performed as follows: 100% A held for 1 min, 100% A to 0% A in 5 min, 0% A held for 2.9 min. Flow rate was set at 0.3 mL/min. Mass spectrometric parameters were set as follows: capillary voltage, 3.80 kV; desolvation temperature, 450 °C and desolvation gas flow, 500 L/Hr. Quantification ion pairs for ToCP and EHDPP (internal standard) were $m/z$ 369 to >164.94, and $m/z$ 363 to >251. Qualification ion pairs for ToCP and EHDPP were $m/z$ 369 to >90.8 and $m/z$ 363 to >152, respectively. A matrix-matched standard curve made with E3 containing 0.1% DMSO was used for quantification. The limit of detection and the limit of quantification were 0.020 μg/L (signal-to-noise ratio = 3) and 0.064 μg/L (signal-to-noise ratio = 10), respectively.

**Data analysis and statistics.** One-way analysis of variance (ANOVA) followed by post-hoc LSD test in SPSS (Chicago, IL, USA) was used to analyze the significance of the differences between the solvent control and the treatments in the developmental parameters of zebrafish embryos. Before running the ANOVA, normality and homogeneity of variances were checked. Results were given as mean $\pm$ standard deviation (SD), and differences were considered as significant at $p < 0.05$.

## 3. Results and Discussion

**Measured ToCP concentrations.** During the 120 h exposure period, two groups of water samples were collected at day 1 and day 3, respectively, for the control and each ToCP treatment. For each group of water samples, equal amounts of water from each replicate were pooled. Additionally, water samples collected at the beginning of the daily experiment (T0) and 24 h later (T24) just before water renewal were mixed to create a composite sample. Water samples were then centrifuged, and the supernatants were collected for UPLC-MS/MS analysis. The mean measured concentrations were recorded as 0.15, 0.51, 3.04, 14.5 and 88.5 μg/L, respectively (Table 1), and were moderately lower than their nominal concentrations of 0.32, 1.6, 8.0, 40 and 200 μg/L, with the decreases ranging from 53.1% to 68.1%. This phenomenon was consistent with the previous reports, where various OPFRs, including TCP and triphenyl phosphate, exhibited significant declines during exposures. Notably, triphenyl phosphate showed reductions of even more than 90% during both short-term (48 h) and long-term (100 days) fish exposures [29,30]. A series of factors could be responsible for this decline, such as rapid photodegradation, hydrolytic degradation, adsorption to particulates and the metabolism by fish, as described [18,29,30].

**Table 1.** Tri-o-cresyl phosphate concentrations (μg/L) in the control and five treatments. Two groups of water samples (collected at day 1 and day 3, respectively) were analyzed during the exposure experiment. For each group, equal amounts of water samples from each treatment, collected at the beginning and after 24 h exposure prior to water renewal, respectively, were mixed. Means of the samples and S.D. are given.

| Nominal Conc. | Control | 0.32 | 1.6 | 8 | 40 | 200 |
|:---:|:---:|:---:|:---:|:---:|:---:|:---:|
| Day 1 | <LOQ | 0.146 | 0.47 | 2.51 | 15.1 | 89.4 |
| Day 3 | <LOQ | 0.154 | 0.55 | 3.58 | 13.8 | 87.6 |
| Mean Conc. | <LOQ | $0.15 \pm 0.01$ | $0.51 \pm 0.06$ | $3.04 \pm 0.78$ | $14.5 \pm 0.9$ | $88.5 \pm 1.3$ |

**Early development of embryos.** Yolk extension, an easily visible trait (specific morphology of yolk extension is displayed in Figure 1A), is a general indicator of early development of zebrafish [26]. A noticeable and significant change in the shape of the yolk cell was observed in the ventral region at the posterior of the 16 hpf zebrafish embryo. This region exhibited a thin and cylindrical shape, which is referred to as the "yolk extension" [26]. As shown in Figure 1A, we found a remarkable acceleration of yolk extension formation in fish embryos. The percentage of yolk extension formation, representing the proportion of embryos displaying such alteration (yolk extension formation as shown in Figure 1A), increased by 4.6%, 8.3%, 4.6%, 48.3% and 128.7% ($p < 0.01$) at concentrations of 0.15, 0.51, 3.04, 14.5 and 88.5 μg/L, respectively. Furthermore, alterations in the spontaneous contraction of zebrafish embryos were also observed. In comparison to the control group, the frequency of spontaneous contractions significantly decreased to 3.8 times per minute (a 15.6% decrease) at a ToCP concentration of 88.5 μg/L ($p < 0.01$), as opposed to the control group's 4.5 times per minute. Spontaneous contraction represents the first sign of locomotion and can be influenced by multiple pathways, including motoneuron axon development and muscle activity [31]. In addition, we also observed a significant change in DEE. As shown in Figure 1C, the DEE of embryos decreased by 1.7%, 3.7% and 5.8% ($p < 0.01$) in the treatment groups, with 3.04, 14.5 and 88.5 μg/L ToCP exposures, respectively, indicating a faster developmental process. Tris (2-chloroethyl) phosphate, one of classic OPFRs, could also accelerate the growth of aquatic organisms, such as *Daphnia magna*, during the early stage of development [32], which is consistent with the present results.

**Cardiac Function.** During the early development of zebrafish, the heart is one of the first organs to form and become functional in the embryo. It plays a critical role in supplying oxygen and nutrients to the entire organism through the vascular system [33,34]. Thus, the proper formation and functioning of the heart are essential for the survival and development of the embryo. We measured the parameters of cardiac function, including heart rate, stroke volume, cardiac output and FAC in 56 hpf zebrafish embryos. As shown in Figure 2, no remarkable effect on heart rate (Figure 1A) and FAC (Figure 1D) was found. Stroke volume increased by 17.8%, 30.0%, 36.2%, 33.2% and 36.7% in the ToCP treatments of 0.15, 0.51, 3.04, 14.5 and 88.5 μg/L, respectively. A significant increase was observed in the 14.5 μg/L exposure group. Similar effects were also observed for triphenyl phosphate. This led to a significant increase in the distance between sinus venosus and bulbus arteriosus (SV-BA distance) of fish embryos at 100 μg/L [35]. In addition, ToCP had a remarkable effect on cardiac output. It led to increases in the cardiac output of 141.5%, 139.6% and 140.8% in the 3.04 μg/L and 14.5 μg/L and 88.5 μg/L treatment group (Figure 2C). Cardiac output, a crucial cardiovascular parameter, measures the volume of blood pumped by the heart per minute, and is estimated by multiplying the stroke volume (SV) with the heart rate. The alteration in cardiac output is not solely determined by the heart's intrinsic factors, but is also influenced by extra factors, such as environmental substances [36]. OPFRs have been observed to accumulate and act in various organs, including the heart, upon chronic exposure [37]. Thus, the present data indicate that the heart might be a potential toxicity target of ToCP.

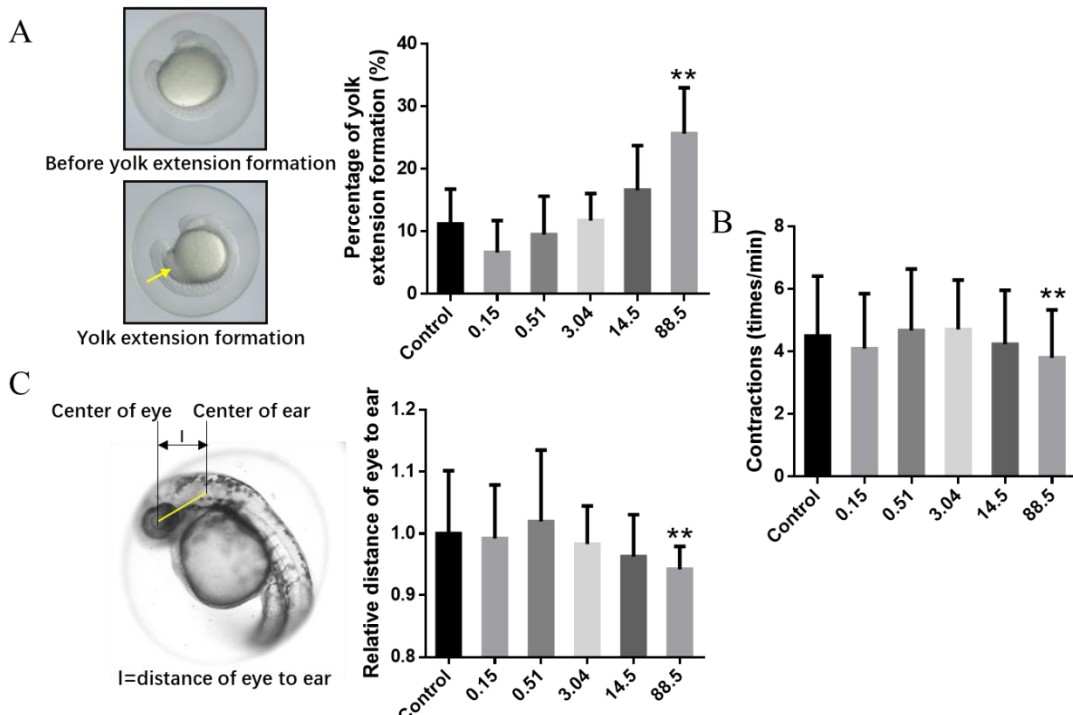

**Figure 1.** Early developmental effects of embryos exposed to ToCP (μg/L). (**A**) Proportion of embryos with yolk extension formation at 16 hpf. (**B**) Spontaneous contractions of embryos at 24 hpf. (**C**) Relative distance from eye to ear of embryos at 36 hpf. $n = 4$ replicates; each replicate contains eight embryos. Asterisks indicate a significant difference compared to solvent control (** $p < 0.01$).

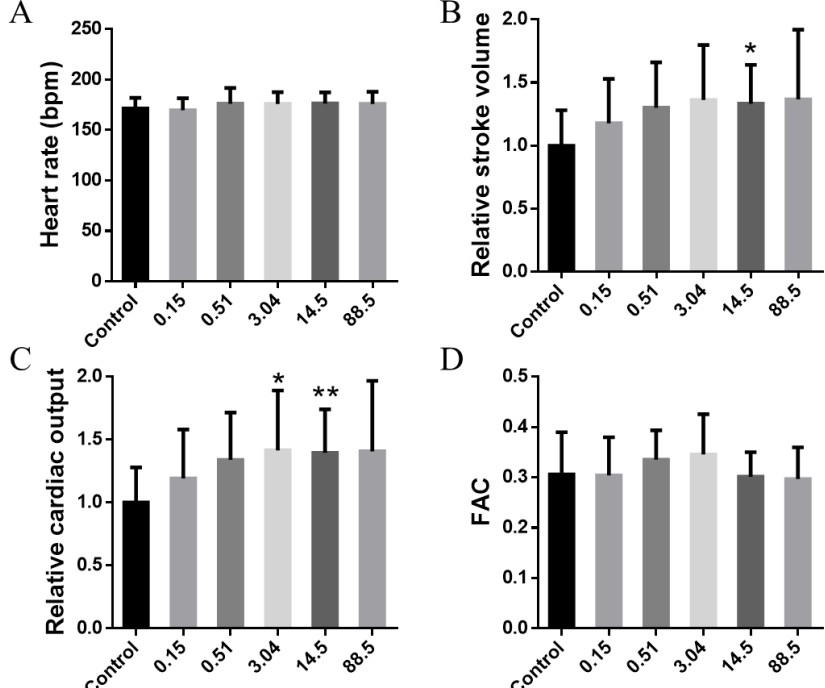

**Figure 2.** Impacts on cardiac function of 56 hpf embryos exposed to ToCP. (**A**) Response of the heart rate. (**B**) Relative stroke volume. (**C**) Cardiac output. (**D**) The ventricular fractional area changes. $n = 4$ replicates; each replicate contains six embryos. Asterisks indicate a significant difference compared to solvent control (* $p < 0.05$, ** $p < 0.01$).

**Body length.** Body length among treatments are displayed in Figure 3. Compared to the control group, a significant decrease of 2.0% was observed at a ToCP concentration of 88.5 µg/L. A similar phenomenon was also observed for other OPFRs, such as triphenyl phosphate and tris(1,3-dichloropropyl) phosphate, occurring at higher concentrations. For example, tris(1,3-dichloropropyl) phosphate led to a reduction in the body length of zebrafish larvae at concentrations >300 µg/L [38,39]. Triphenyl phosphate significantly reduced the body length of zebrafish larvae at a concentration of 100 µg/L [40]. In addition, both triphenyl phosphate and tris(1,3-dichloropropyl) phosphate significantly decreased the body length and affected the development of *Daphnia magna* in a dose-dependent manner [41]. A smaller body size could be indicative of slower development of embryos, which may lead to differences in various developmental parameters. Thus, the decreases in body size observed in this study could potentially influence and contribute to alterations in spontaneous contraction behavior and eye-to-ear distance as described above.

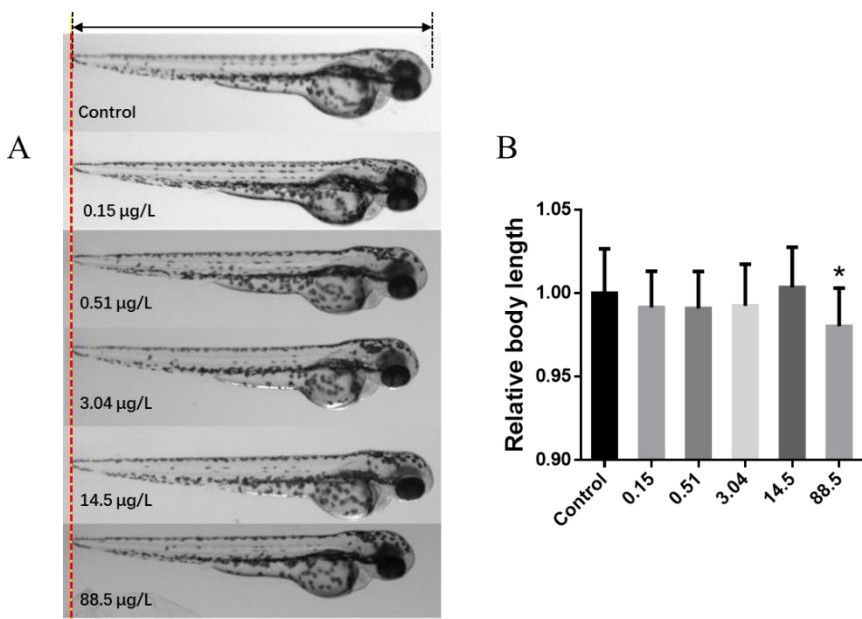

**Figure 3.** Alterations of the body length of embryos exposed to ToCP. (**A**) Graphs and (**B**) bar figures represent the changes in body length of zebrafish embryos at 60 hpf. *n* = 4 replicates; each replicate contains eight embryos. Asterisk indicates a significant difference compared to solvent control (* *p* < 0.05).

**Swimming Behavior.** The swimming activity of 120 hpf zebrafish embryos was further assessed to investigate potential neurobehavioral effects of ToCP. As shown in Figure 4, ToCP exposure led to remarkable inhibition of the swimming behavior. Compared to the solvent control, the total swimming distance, swimming velocity and swimming time all decreased dose-dependently. In the light phase, ToCP led to significant decreases in swimming distance of 19.5% (*p* < 0.01) and 29.2% (*p* < 0.001) at 14.5 and 88.5 µg/L, respectively (Figure 4A). In the dark phase, it led to significant decreases of 7.7% (*p* < 0.05), 9.7% (*p* < 0.05) and 16.9% (*p* < 0.001) at 3.04, 14.5 and 88.5 µg/L, respectively (Figure 4D). Similarly, ToCP significantly decreased the light-phase swimming velocity by 19.5% (*p* < 0.01) and 29.2% (*p* < 0.001) at 14.5 and 88.5 µg/L, respectively (Figure 4B). In the dark phase, it led to significant decreases in swimming velocity of 7.7% (*p* < 0.05), 9.7% (*p* < 0.05) and 16.9% (*p* < 0.001) at 3.04, 14.5 and 88.5 µg/L, respectively (Figure 4E). Such phenomena can also be found for the swimming time, as shown in Figure 4C,F. Thus, the present data clearly demonstrate adverse effects of ToCP on zebrafish swimming activities. These effects were observed at ToCP concentrations ranging from 3.04 to 88.5 µg/L. ToCP residues have been detected in domestic wastewaters at levels of up to 0.56 µg/L [9].

These concentrations are comparable to those tested in the present study; thus, these findings may indicate the potential relevance and implications of ToCP exposure in natural water bodies. In should also be noted that the observed responses in swimming activity were more pronounced in dark conditions compared to light conditions. This observation aligns with previous reports, where zebrafish swimming activity was significantly affected by environmental substances, such as paraquat and acenaphthene, under dark conditions, while minimal changes occurred in light conditions [42,43]. This difference in response may be attributed to the naturally higher swimming activity levels of zebrafish embryos in dark conditions. Therefore, this phenomenon warrants careful consideration in future chemical risk assessment.

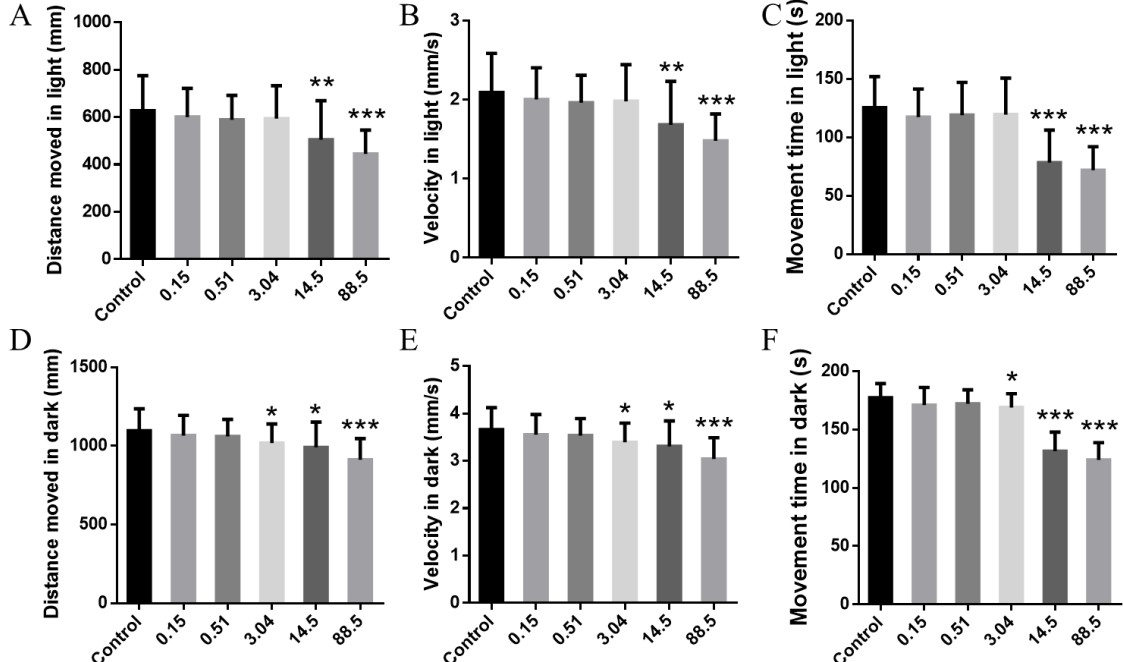

**Figure 4.** Alterations of swimming behavior of embryos upon ToCP exposure. (**A–C**) Changes in swimming distance, swimming velocity and swimming time in the phase of light. (**D–F**) Changes in swimming distance, swimming velocity and swimming time in the phase of dark. *n* = 4 replicates; each replicate contains six embryos. Asterisks indicate a significant difference compared to solvent control (* $p < 0.05$, ** $p < 0.01$, *** $p < 0.001$).

Furthermore, we measured the swimming activity during the light-to-dark transition. We defined the total movement distance in the ten seconds before the light turning off as $L_{light}$ and defined the total movement distance in the ten seconds after the light turning off as $L_{dark}$. The swimming distance during transition sets was defined as $\Delta L = L_{dark} - L_{light}$. As shown in Figure 5, we observed significant increases in swimming distances $\Delta L$ of 38.3% ($p < 0.01$) and 53.8% ($p < 0.001$) in embryos upon 14.5 and 88.5 µg/L ToCP exposure, respectively. In investigations pertaining to zebrafish neurobehavioral toxicity, behavioral change during the light-to-dark transition serves as a widely recognized parameter that stands as an important determinant of the light responsiveness exhibited by fish embryos, encompassing their ability to perceive and respond to light stimuli effectively [44]. Therefore, the behavioral changes observed here may hold important value in comprehending the responsiveness of embryos to light stimuli upon ToCP exposure.

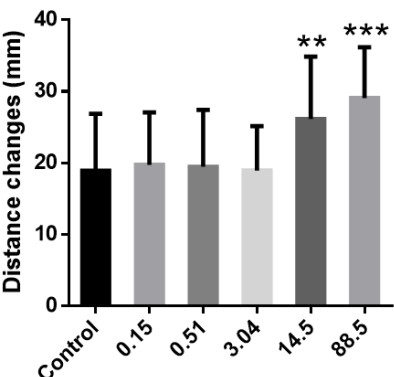

**Figure 5.** Changes in swimming distance of 120 hpf embryos during light-to-dark transition. *n* = 4 replicates; each replicate contains six embryos. Asterisks indicate a significant difference compared to solvent control (** $p < 0.01$, *** $p < 0.001$).

Taken together, our data illustrated that ToCP might act as a neurobehavioral toxicant to zebrafish embryos at relative low concentrations (3.04–88.5 µg/L). As a comparison, reports have demonstrated that some OPFRs, such as 2-ethylhexyl diphenyl phosphate, butylphenyl diphenyl phosphate and isodecyl diphenyl phosphate, also led to inhibitions of the swimming activity of zebrafish. However, such effects generally occurred at mg/L levels [45,46].

Thus, the findings of this study indicated significant developmental effects of ToCP on zebrafish embryos, particularly affecting their swimming activities. These alterations may have far-reaching implications for survival, social dynamics and even reproductive behaviors in adult fish. Meanwhile, the impacts observed in zebrafish could potentially extend to other fish species within the aquatic ecosystem, which deserves thorough investigation. It should be noted that the present study also acknowledges limitations that warrant consideration. For instance, a significant inhibitory effect at a ToCP concentration of 88.5 µg/L was observed for spontaneous contraction, while the assessment was limited to a single time point (24 hpf). Given the complexity of spontaneous contraction in zebrafish embryos that evolves over time, a comprehensive approach involving multiple time points is essential for yielding a valuable insight into such developmental impacts. Furthermore, this study deviated from the standard Fish Embryo Toxicity (FET) test according to OECD TG 236. The OECD TG 236 test is a valuable tool for assessing the effects of acute chemical toxicity on the embryonic stages of fish, while our research aimed to gain an in-depth understanding of the early developmental toxicities of ToCP by examining a wider range of parameters, which encompassed a broader range of toxic effects that went beyond the scope of acute toxic effects. As a result, our findings have unveiled remarkable toxic effects of ToCP, which require further mechanistic investigations.

In conclusion, the present study demonstrated that ToCP, a major isomer of TCP, could lead to significant toxic effects on the early development of zebrafish embryos, encompassing neurobehavioral toxicity and cardiac dysfunctions. Particularly, the alterations of swimming behaviors occurred at ToCP concentrations ranging from 3.0 µg/L to 88.5 µg/L. Given that residues of ToCP have been detected in domestic wastewater at levels up to 0.56 µg/L and in surface water at levels up to 0.11 µg/L [9,11], which are comparable to those tested in the present study, the present data may underline the potential relevance and implications of ToCP exposure in natural water bodies. Furthermore, the results provide us with essential insights into the impacts of ToCP on zebrafish, but may also be important for other fish species and aquatic organisms, as OPFRs such as TCP have already been detected in various biota samples, such as tongue sole and octopus [47,48].

**Author Contributions:** M.L.: Investigation, Methodology, Data analysis, Visualization, Writing-original draft.; C.W., W.G. and J.C.: Investigation, Methodology; S.Z. and P.W.: Investigation, Methodology, Data analysis; Y.Z. and J.D.: Writing-review & editing; K.Z.: Supervision, Conceptualization, Funding acquisition, Writing-review & editing. All authors have read and agreed to the published version of the manuscript.

**Funding:** This research is supported by the National Natural Science Foundation of China [21976121, 22122605 and 22006099] and the National Key Research and Development Program of China [2022YFC2705200].

**Conflicts of Interest:** The authors declare no competing financial interest.

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
