# Peer review of "Developmental Toxicities in Zebrafish Embryos Exposed to Tri-o-cresyl Phosphate"

_water, doi:10.3390/w15162942_

Round 1
Reviewer 1 Report
The publication aimed to determine the developmental toxicity of tri-o-cresyl phosphate on zebrafish embryos, to fill the current knowledge gap. The indoduction highlights the need for this data well, stating different environmental concentrations and placing the subject in the wider context. I question the environmental relevance this study claims to have, however, as the concentrations chosen for the exposure are, in fact only barely touching environmental concentrations, and as such not informing the stated aim of the study. It would be more suitable completely remove the claim of environmental relevance. I am not confident in the selection of timepoints for some of the observations. It could be discussed that yolk extension is a valid observation even without assessing yolk resorption later on, but assessing coiling behaviour (or spontaneous contraction as called here) for only 30 secs early on in the development of the coiling behaviour is a grave mistake if the intention is to understand behavioural impacts. Many other standard developmental endpoints, as stated in the OECD TG 236, were also completely ignored. For the claims made in the results and discussion, as well as conclusion, these endpoinds must be assessed in more detail than done here. However, the swimming data presented here is great and should be the real focal point of the paper, should it be accepted for publication.
Line 1: Consider shortening the title e.g., "Developmental toxicity in zebrafish embryo exposed to sub-acute tri-o-cresyl phsophate concentration
Line 9: Please insert a space betweeen ":" and "Dr. Kun Zhang ..."
Line 11: This sentence is missing information. Consider ending sentence with "...is not fully understood."
Line 12: What is meant with systematically assessing the effects?
Lines 14-18: I suggest shortening this section, stating that "Diverse impairments, such as morphological development and locomotor behaviour, were observed at different concentrations."
Line 19: No need to define the extent of swimming behaviour you assessed. I suggest only higlighting the finding
Line 25: I suggest you replace all keywords other than "swimming behaviour" with words that are not already in the title. The keywords and title help search engine optimisation and should thus not be the same
Line 30: Replace "have" with "has"
Line 43: Remove "in total"
Line 45: You reference 54 ng/l in surface water and 107 ng/l in drinking water... The exposure concentrations you are using are far beyond this, and of no environmental relevance? Please explain why you chose that concentration range and what you hope to determine here
Lines 49-52: Please convert the high ng/g values to ug/g
Lines 72-73: please convert uM to ug or equivalent, to improve the comparability of the findings you are refering to
Lines 80-82: This is not a sentence for an aims/objectives text. Please einther re-write or place in the conclusion
Section 2.2: Is there a reason why you did not do a standard FET test (OECD TG 236)? That would allow you to compare your data globally and gain further insights about the developmental toxicity of the compound
Line 102: This is incorrect, unless you did not cite the correct values. Reference 9 states 560 ng/l in domestic wastewater. That is 0.56 ug/l, which is only in the range of the lowest concentration you selected. With this approach, you cannot gain insights into the potential effects at the selected environmental concentration, as you have no ... Not to forget that wastewater is not the environmental norm, as it will be diluted in the receiving body of water, etc. Please clarify why you feel that your selected concentration range is environmentally relevant
Line 104: I assume the medium was exchanged daily, but that is never stated here. Please rectify
Lines 113-121: Font size changes here. Please be careful when copying in sections
Sections 2.4-2.6: I suggest placing this before section 2.3 as these are method for the ZFE treatment
Line 129: What is the reason for not observing yolk resorption in 120h old ZFE? This is highlighly informative for development
Lines 129-132: Coiling really needs to be observed for multiple time points, not just at 24h. The behaviour develops over time and changes could occur at any point along that development!
Line 145: Please state the hour age of the larvae used here
Section 2.7: Why was no EC or LC value computed?
Table 1: I am not clear which sample was taken when, and why you did not take samples from the same medium (i.e. one fresh and one just before medium renewal). This also means that it is pointless to calculate the mean concentration, considering that you are looking at pre- and post-uptake concentrations? Also, I understand how factors lead to reduced actual concentrations in later samples, but how can relatively fresh medium of Day 1 already show concentration decreases of more then 50%?
Lines 178-179: Please explain what the percentages mean? Is the the amount of additional yolk formed, or the percentage of exposed animals with the alterations?
Lines 180-185: This is not clear from your results at all. Longer tim-series analyses are needed to make any such claim, especially considering the relatively small nr of indviduals tested and the high standard deviation in all concentrations. Also be careful with selling uncertain results as clearly "demonstrating" something. there is no "strong impact on motor neurogenesis" and the compound could affect any number of pathways to lead to reduced coiling. E.g. muscle or overall development
Section 3.3: Again, there is no clear concentration dependent trend, as there is a high standard deviation in all datasets and the significance is only dotted around two parameters. It does not "clearly demonstrate" anything
Section 3.4: This is the first time you mention body length measurements. Please add method into the respective section
Line 218: This could already be the reason for changes in the coiling behaviour and eye-to-ear distance. Smaller body could mean less fast developed and thus not yet where cluch-related embryos are in other exposure groups
Section 3.5: This is great data! I would aim to focus the study on this, as you have some novel data and great insights here
Lines 264-272: Please rewrite this to not oversell the data presented here. Your swimming results show that there is a definite impact on late developmental behaviour, which COULD impact survival and social groups in adult ZF. However, without further work looking at other endpoints during development (OECD TG 236...) and a "proper" coiling assay (e.g. Zindler et al., 2019-2020) the other insights must be considered tentatively!
The overall language of the manuscript is well worded, but often the wrong tense of a verb is used, or the sentence is so overly complicated that it looses its readability. It is advisable to re-read the manuscript from that standpoint, or to request an additional review by a native speaker not invlolved with the manuscript already.
Reviewer 2 Report
1. In the paper, you talk about the software "ImageJ" and refer to the "previously developed method" for the interpretation of experimental data. I do not identify the presentation of the method in the paper and therefore I ask you to present it in order to complete its content.
2. The results recorded by the research group are conclusive and they adequately show that the studied pollutant has significant negative effects on the analyzed fish species. It is possible that it also has negative effects on all fish species, including the entire underwater environment.
3. The conclusions presented by the authors are irrelevant for the scope and importance of the results obtained and therefore I recommend their expansion so that they offer the reader a relevant synthesis of the results according to the parameters used in the experiments and reveal a complete picture of the need to protect surface water bodies.
Reviewer 3 Report
The paper is relatively well written and addresses an important topic. A careful proofreading or professional editing is needed to improve the presentation of the paper. There are many grammar errors and awkward sentences. E.g., p.1, ln.11-12: “its potential toxicity to aquatic organisms is far from complete.”; p.1, ln.41: “it consists”; p.1,ln.44-45: “found by”; p.2, ln.74-5: “ thorough understanding of the potential environmental risks of ToCP on
aquatic organisms are still lacking”; p.2, ln.90: “to the experiment”; some texts have different font sizes than others (e.g., p.2, ln.109-10); p.4, ln.185: “strong impacts”; p.4, ln.186: “decrease”; p.5, ln.199: “no remarkably effect”; p.5, ln.202: “significantly increase”.
The paper is relatively well written and addresses an important topic. A careful proofreading or professional editing is needed to improve the presentation of the paper. There are many grammar errors and awkward sentences. E.g., p.1, ln.11-12: “its potential toxicity to aquatic organisms is far from complete.”; p.1, ln.41: “it consists”; p.1,ln.44-45: “found by”; p.2, ln.74-5: “ thorough understanding of the potential environmental risks of ToCP on
aquatic organisms are still lacking”; p.2, ln.90: “to the experiment”; some texts have different font sizes than others (e.g., p.2, ln.109-10); p.4, ln.185: “strong impacts”; p.4, ln.186: “decrease”; p.5, ln.199: “no remarkably effect”; p.5, ln.202: “significantly increase”.
Reviewer 4 Report
The manuscript of Li et al entitled “Organophosphate Ester Tri-o-cresyl phosphate Induces Developmental Toxicities in Zebrafish Embryos at Low Concentrations” shows the potential toxic effect of Tri-o-cresyl in Zebrafish, particularly in Neurobehavioral activity and cardiac function. Results found by the authors is particularly important due to the potential threat of such compound not only to aquatic organisms but also to human health. I suggest the authors to minor considerations:
1. In lines 160-161, the authors say “Two groups of water samples … were analyzed during the exposure experiment.” In this sentence, inform which analysis was accomplished.
2. In lines 266-267, “The effects occurred at ToCP concentrations of several μg/L and above”: I suggest including at least a range of these “several μg/L and above”. Furthermore, after this sentence, the authors say “These data are of significance considering that the residues of ToCP in surface waters have reached up to hundreds of ng/L”. I suggest authors use the same unit (ug/L) used in the previous sentence to highlight the comparison. In addition, in “conclusions’, it would be interesting to comment again, on what was said in lines 102-103: ToCP concentrations that represent environmentally relevant doses in surface waters already cause serious effects on the embryonic development of zebrafish.
Round 2
Reviewer 1 Report
Dear authors, I appreciate the amount of work that will have gone into incorporating my suggestions (as well as those of other reviewers). This is an interesting piece of work and I hope that it will get a large readership once published.
Other than a few very minute aspects, I see no issue with the language and don't feel that the manuscript needs any further work.